# Using Multivariate Adaptive Regression Splines to Estimate the Body Weight of Savanna Goats

**DOI:** 10.3390/ani13071146

**Published:** 2023-03-24

**Authors:** Lebo Trudy Rashijane, Kwena Mokoena, Thobela Louis Tyasi

**Affiliations:** 1Department of Agricultural Economics and Animal Production, School of Agricultural and Environmental Sciences, University of Limpopo, Private Bag X1106, Sovenga, Polokwane 0727, Limpopo, South Africa; 2Agricultural Research Council, Biotechnology Platform, Private Bag X5, Onderstepoort, Pretoria 0110, Gauteng, South Africa

**Keywords:** data mining algorithm, linear body measurements, goodness of fit, correlation

## Abstract

**Simple Summary:**

The current study was conducted to predict the live body weight using body length, heart girth, rump height and withers height of 173 Savanna goats from a stud breeder at Bysteel, Polokwane municipality, South Africa. A multivariate adaptive regression splines algorithm was used, along with the different proportions of the test and training sets to predict body weight. The body weight was best predicted from the training dataset with body weight influenced by withers height and heart girth, respectively. The interaction of withers height and body length with withers height and heart girth also influenced body weight. In conclusion, it could be suggested that the multivariate adaptive regression splines algorithm might allow Savanna goat breeders to find the best population and examine the body measurements affecting body weight as indirect selection criteria for describing the breed description of Savanna goats and aiding sustainable meat production.

**Abstract:**

The Savanna goat breed is an indigenous goat breed in South Africa that is reared for meat production. Live body weight is an important tool for livestock management, selection and feeding. The use of multivariate adaptive regression splines (MARS) to predict the live body weight of Savanna goats remains poorly understood. The study was conducted to investigate the influence of linear body measurements on the body weight of Savanna goats using MARS. In total, 173 Savanna goats between the ages of two and five years were used to collect body weight (BW), body length (BL), heart girth (HG), rump height (RH) and withers height (WH). MARS was used as a data mining algorithm for data analysis. The best predictive model was achieved from the training dataset with the highest coefficient of determination and Pearson’s correlation coefficient (0.959 and 0.961), respectively. BW was influenced positively when WH > 63 cm and HG >100 cm with a coefficient of 0.51 and 2.71, respectively. The interaction of WH > 63 cm and BL < 75 cm, WH < 68 cm and HG < 100 cm with a coefficient of 0.28 and 0.02 had a positive influence on Savanna goat BW, while male goats had a negative influence (−4.57). The findings of the study suggest that MARS can be used to estimate the BW in Savanna goats. This finding will be helpful to farmers in the selection of breeding stock and precision in the day-to-day activities such as feeding, marketing and veterinary services.

## 1. Introduction

The Savanna goat breed is one of the South African indigenous goat breeds that is reared for meat production [1,2]. This breed was primarily chosen for its excellent fertility, ease of care and tolerance to heat and drought [3]. According to Stonehavenstud [4], this breed is commonly known for its rapid growth, moderately high milk production and excellent mothering ability. Mohlatlole et al. [5] highlighted that this breed has a large body frame and underwent strict selection for larger carcasses and rapid growth. The live body weight of an animal is very crucial since it helps farmers manage their livestock during selection, feeding and medical dosages [6,7,8]. Therefore, in situations where weighing equipment is unavailable, body weight prediction in farm animals is used as a perfect alternative [7]. The prediction of body weight and its association with linear body measurements is also important for improving body weight during breeding using linear body measurements [9]. 

Numerous studies were conducted on the estimation of body weight from linear body measurements in goats using regression techniques [10,11,12]. However, Eyduran et al. [10] indicated that the regression techniques cannot overcome the multi-collinearity problems from the variables. Hence, some studies used different data mining algorithms such as the classification and regression trees (CART) in South African Boer goats [6], chi-squared automatic interaction detector (CHAID) in South African indigenous non-descript goats [12], exhaustive CHAID in Kalahari Red goats [13] and multivariate adaptive regression splines (MARS) in Pakistani goats [9] for more and better developmental breeding strategies. Eyduran et al. [14] indicated that these data mining algorithms were also useful for the estimation of fleece weight from the wool characteristics of sheep. MARS data mining algorithm is a type of regression analysis technique [15], and it was used to develop the prediction equation for body weight using linear body measurements [7]. According to Ige et al. [16], MARS is the best data mining algorithm for the development of improved animal breeding strategies. It was noted that this data mining algorithm could overcome the multi-collinearity problems from the estimation of body weight using linear body measurements [17,18]. Hlokoe et al. [19] compared the MARS and CART on the estimation of the live body weight of Nguni cows and suggested that MARS was a suitable statistical tool that can be used to describe breed standards for breeding. However, there is limited information on these data mining algorithms [10] for the estimation of body weight. 

Still, based on the attained information, arguably, there is limited knowledge of the usage of MARS to predict the effect of linear body measurements on the body weight of Savanna goats. Hence, the main purpose of the current study was to estimate the live body weight of Savanna goats from some linear body measurements through multivariate adaptive regression splines. The other goal of the current study was to determine the traits which can be used by goat farmers to determine body weight without the use of a weighing scale. Data mining algorithms were used to determine the best traits which can be used to estimate body weight. 

## 2. Materials and Methods

### 2.1. Study Area and Animal Management

The study was conducted at the game breeder farm in Bysteel, Polokwane municipality, South Africa. This place was reported to experience an annual rainfall of 600 mm and more, with summer temperatures ranging from 16 °C to 28.1 °C and winter temperatures ranging from 7 °C to 21 °C [20]. A total of 173 Savanna goats (16 bucks and 157 does) between the ages of 2 and 5 years were used in this study. The goats were reared under an extensive management system. They were given the privilege to graze what was available on the farm during the day and return to the kraal before sunset, where they were monitored and provided with clean water. During this study, only healthy and non-pregnant animals were used for accuracy. The animals were chosen randomly. 

### 2.2. Data Collection

Body weight (BW), heart girth (HG), rump height (RH) and withers height (WH) were measured during this study. Body weight was measured in kilogrammes using a weighing scale [21]. The body measurement traits were collected using a wood ruler and measuring tape in centimetres following the procedure described by Tyasi et al. [22]. Briefly, HG was measured as the distance from the body circumference at a position immediately posterior to the front leg and shoulder and perpendicular to the body axis. RH was measured as the space from the surface of a platform to the rump. WH was measured as a vertical position between the ground and the apex of the tourniquet, immediately behind the hump and on the top of the scapula. All the measurements were taken by one individual to avoid individual variation in measurements. 

### 2.3. Multivariate Adaptive Regression Spline (MARS) Algorithm

MARS is a non-parametric regression method developed by Friedman [23]. In this study, the MARS algorithm was conducted as explained by Şengül et al. [24], and its prediction equation can be written as follows:$$f\left(x\right)=\beta_{0}+\overset{m}{\underset{m=1}{\sum }}\beta_{m}\lambda_{m}\left(x\right)$$
where *f*(*x*) is the expected response, $\beta_{0}$ and $\beta_{m}$ are parameters that are calculated to give the best data fit, and m is the number of BFs in the model. In the MARS model, the basis function composed of be a single univariable spline function or a combination of more than one spline function for diverse predictor inputs. The spline BF, $\lambda_{m}\left(x\right)$, is defined as: $$\lambda_{m}\left(x\right)=\overset{k_{m}}{\underset{k=1}{\prod }}\left[s_{km}\left(X_{v\left(k,m\right)}-t_{k,m}\right)\right]\;$$
where $t_{k,m}$ denotes the knot location; $s_{km}$ denotes the right/left regions of the corresponding step function, taking either 1 or −1; v(k,m) denotes the predictor variable’s label; and *k_m_* is the number of knots. Following the procedure of Şengül et al. [24], the pruning process was used to remove the basic functions that had a low contribution to the model fitting performance following the generalised cross-validation error (GCV):$$GCV\left(\lambda \right)=\frac{\sum_{i=1}^{n}{\left(y_{i}-y_{ip}\right)}^{2}\;}{{\left(1-\frac{M\left(\lambda \right)}{n}\right)}^{2}}$$
where *n* represents the number of training cases, $y_{i}$ shows the observed value of the responsible variable, $y_{ip}$ as the estimated value of the response variable and *M*(λ) represents the penalty function for the complex of the model with λ terms.

The following goodness of fit test criteria were computed for training and test datasets: 

Pearson’s correlation coefficient (r):$$r=\frac{\mathrm{cov}\left(y_{i},y_{\mathrm{ip}}\right)}{S_{\mathrm{yi}\;}S_{\mathrm{Yip}}}$$

Relative root mean square error (RRMSE):$$\mathrm{RRMSE}=\frac{\sqrt{\frac{1}{n}\sum_{i=1\;}^{n}(y_{i}-y)\times 2}}{\bar{y}}$$

Mean error (ME):$$\mathrm{ME}=\frac{1}{n}\overset{n}{\underset{i=1}{\sum }}\left(y_{i}-y_{\mathrm{ip}}\right)$$

Performance index (PI):$$\mathrm{PI}=\frac{\mathrm{rRMSE}}{1+r}$$

Coefficient of determination (Rsq): $$\mathrm{Rsq}=1-\frac{\sum_{i=1}^{n}{\left(y_{i}-{\hat{y}}_{i}\right)}^{2}}{\sum_{i=1}^{n}{\left(y_{i}-\bar{y}\right)}^{2}}$$

Coefficient of determination is used to measure the proportion of variation explained by the independent variables for the dependent variable, where r represents the correlation coefficient between the fitted and observed body weight. 

Adjusted coefficient of determination (ARsq):$$\mathrm{ARseq}=1-\frac{\frac{1}{n-k-1}\sum_{i=1}^{n}{\left(y_{i}-{\hat{y}}_{i}\right)}^{2}}{\frac{1}{n-1}\sum_{i=1}^{n}{\left(y_{i}-\bar{y}\right)}^{2}}$$

Root-mean-square error (RMSE):$$\mathrm{RMSE}=\sqrt{\frac{1}{n}\sum_{i=1}^{n}{\left(y_{i}-{\hat{y}}_{i}\right)}^{2}}$$
where n shows the total number of samples used; y_i_ and ${\hat{y}}_{i}$ represent the observed and fitted weights of the ith animal, respectively.

Standard deviation ratio (SDR):

This is an evaluation measure that is used in assessing the performance of fitted models by taking the ratio of the observed to the fitted model’s values.
$$SD_{Ratio}=\sqrt{\frac{\frac{1}{n-1}\sum_{i=1}^{n}{\left(\varepsilon_{i}-\bar{\varepsilon }\right)}^{2}}{\frac{1}{n-1}\sum_{i=1}^{n}{\left(Y_{i}-\bar{Y}\right)}^{2}}}$$

Akaike information criteria (AIC):

The method, AIC, is used in evaluating how good a model fits the data. It is used to choose the best for the data by comparing its fit to the data.
$$\mathrm{AIC}=N\;\mathrm{Ln}\;\left(\frac{\mathrm{SSE}}{N}\right)+2p$$

Mean absolute percentage error (MAPE): 

MAPE is another popular measure used to predict error. It is easy to understand and interpret as it measures the size of the error in percentage terms.
$$\mathrm{MAPE}=\frac{1}{n}\sum_{i=1}^{n}\left|\frac{Yi-\hat{Y}i}{Yi}\right|\;\times \;100$$

Mean absolute deviation (MAD):

MAD is used to avoid the issues of negative and positive errors cancelling each other out from the MAE. The smaller the MAD, the better the fit.
$$\mathrm{MAD}=\frac{1}{n}\sum_{i=1}^{n}\left|\frac{Yi-\hat{Y}i}{Yi}\right|$$

Global relative approximation error (RAE): $$\mathrm{RAE}=\sqrt{\frac{\sum_{i=1}^{n}({Yi-\hat{Y}i)}^{2}}{\sum_{i=1}^{n}Y\frac{2}{i}}}$$

Coefficient of variance (CV):$$\mathrm{CV}=\sqrt{\frac{\frac{1}{n-1}\;\sum_{i=1}^{n}{\left(\varepsilon_{i}-\;\bar{\varepsilon }\right)}^{2}}{\;\bar{Y}}}\times 100$$
where Yi is the observed live body weight (kg) of ith goats; $\hat{Y}i$ is the predicted live body weight value of the ith goats; $\;\bar{Y}$ is an average of the actual live body weight values of the goats;$\;\varepsilon_{i}$ is the residual value of the ith goats; an average of the residual values; k is the number of significant independent variables in the model, and n is the total number of goats. The residual value of each goat is expressed as $\varepsilon_{i}$ = Yi − $\hat{Y}i$. 

The Savanna goats’ data set was divided into two data sets, training and test sets at proportions 70–30%. In the training set, a ten-fold cross-validation resampling method was used to select the best MARS models with degree = 1:9 and n prune = 2:38 as a number of selected terms within the scope of live body weight estimation. 

Package in R was used in the statistical evaluation of the MARS data mining algorithm for the prediction of body weight. EhoGof package (version 0.1.1, Igdir, Turkiye) developed by Eyduran [25] in R was implemented to reveal the predictive performance of the optimal MARS model.

## 3. Results

Descriptive statistics of BW and linear body measurements for does are presented in Table 1. The BW ranged from 22 kg to 66 kg, while the CV ranged from 7.32% to 24.76%. 

Table 2 represents descriptive statistics for bucks with BW ranging from 34 kg to 102 kg. The results showed 67.50 ± 3.54 cm for WH, 73.75 ± 2.25 cm for RH, 76.623 ± 2.48 cm for BL and 86.88 ± 4.06 cm for HG.

Table 3 shows the correlation coefficient for determining the association between BW and linear body measurements. In bucks, BW had a highly significant correlation (*p* < 0.01) with RH, HG, BL and WH. The results indicated that the highest significant correlation was observed between HG and RH (*p* < 0.01) among linear body measurements. In does, the result showed that BW had a highly significant correlation (*p* < 0.01) with HG, WH, BL and RH. Among linear body measurements, RH and WH had the highest correlation (*p* < 0.01) in does.


**Performance of the MARS model**


The performance for the MARS model results of training and test dataset based on goodness of fit is given in Table 4. The results showed that the best predictive model was achieved from the training dataset for the proportion 70% (Training)–30% (Test). The training set had the lowest RMSE, SD_Ratio_, CV, PI, RAE, MAPE, MAD and r values. The Rsq and Pearson’s correlation values for the training set were higher than those of the test set.

The model (Table 5) established by the MARS data mining algorithm showed that WH, BL, HG and RH were involved in the model. There were eight basic functions established from the MARS model with five single-order term variables and three orders of interaction with an intercept of 53.21. MARS described the influence of linear body measurements with the negative and positive coefficients on BW. Briefly, the influence on BW of Savanna goats was in the positive direction, and the model coefficient was 0.51 when WH > 63 cm, and the model coefficient was 2.71 when HG > 100 cm. Additionally, the model revealed the effect of linear body measurements interaction on BW. The influence on BW was 0.28 when WH > 63 cm and BL < 75 cm. The model coefficient was 0.02 when WH < 68 cm and HG < 100 cm. Bucks (SexM) had a negative effect on BW with a coefficient of −4.57. 

A graph of candidate MARS models tested with the aid of train function in caret R package is depicted in Figure 1. The lowest RMSE (cross-validation) was obtained by the MARS model with four terms and the third-order interaction (degree = 4).

## 4. Discussion

The methods of body weight estimation using body measurements are commonly used in determining the association between the structures of animal species [24]. The phenotypic correlations reported a significant correlation in both bucks and does of the Savanna goats. The body weight showed a significant correlation with body length, heart girth, rump height and withers height in bucks, while in the does, body weight was significantly correlated to heart girth, withers height, body length and rump height. These findings support the study by Lan et al. [26] which showed that heart girth can be used to predict body weight in tropical goat breeds. Furthermore, Abd-Allah et al. [27] reported that linear body measurements are ideal to predict the live body weight in Shami goats, with heart girth significantly predicting body weight in bucks and body length in does. Temoso et al. [28] conducted a study in goats and sheep of Botswana communal rangelands and concluded that the heart girth was the best predictor of body weight in both sheep and goats. Maylinda and Busono [29] reported a strong relationship between body weight with chest circumference and tail circumference in fat-tailed sheep. In broiler chicken, Sadick et al. [30] reported that body weight can be estimated from linear body measurements with the best predictor of body weight indicated for shank circumference.

The current results suggest that there is a relationship between body weight and linear body measurements in Savanna goats. The association observed between the studied traits recommends that they might be controlled by the same gene [12]. Correlation results of the current study imply that improving the withers height, rump height, body length and heart girth of Savanna goats might improve the live body weight. Therefore, these traits can be useful as selection criteria for genetic improvement of live body weight during goat breeding. The correlation coefficient does not provide the effect of linear body measurement traits on body weight. Hence, multivariate adaptive regression splines (MARS) were used to determine the effect of linear body measurements on the body weight of Savanna goats. MARS results showed that sex, withers height, rump height, body length and heart girth were determined as the important parameters for the prediction of live body weight in Savanna goats. Due to the scarcity of MARS data mining algorithm studies in goats, we discussed our MARS findings using different animal species. Ağyar et al. [31] indicated that the MARS results determined tail length, body length, chest circumference and shank diameter as important parameters in Anatolian buffaloes in Turkiye. The MARS results implied that sex, withers height, rump height, body length and heart girth might be useful for the improvement of live body weight in Savanna goats. The MARS results showed the best predictive model was achieved from the dataset with a proportion of 70:30 (70 for the training:30 for the test). The predictive performance of the MARS data mining algorithm of the current study showed that root mean square error, relative root mean square error, standard deviation ratio, coefficient of variation, performance index, mean error, relative approximation error, mean absolute percentage error, mean absolute deviation and alkaike’s information criterion were lower in the training dataset than test, whereas Pearson’s correlation coefficient, coefficient of determination and adjusted coefficient of determination were higher in the training dataset than the test. The Pearson’s correlation coefficient and coefficient of determination for this study were higher than the study by Tirink et al. [32], which highlighted that the model with 80:30 proportions was the best model in the Marercha camel of Pakistan with the MARS model that resulted in eight basic functions. Faraz et al. [33] reported a lower goodness of fit for the prediction of live body weight in Thalli sheep, and Tyasi et al. [34] reported a lower goodness of fit for the prediction of body weight of the Hy-Line Silver Brown commercial layer chicken breed. 

Fatih et al. [35] further explained that MARS can be used to predict live body weight in camel breeds in Pakistan with a higher coefficient of determination and adjusted coefficient of determination (0.95, 0.92) than the current study. The superiority of MARS as a predictor of body weight was reported in Pakistani goats with a lower coefficient of determination and adjusted coefficient of determination than in the current study [9]. Celik and Yilmaz [36] reported a lower coefficient of determination and standard deviation ratio in the prediction of Kars Shepard dogs. In Turkish dogs, the prediction of body weight using MARS recorded a predictive accuracy of 0.6889 and an adjusted coefficient of determination of 0.9193, and these results were lower than the results reported in the current study [37]. Our MARS model performance results implied that MARS might be useful for the prediction of live body weight from linear body measurements in goats using 70:30 dataset proportion for training and test sets. These findings agree with Ağyar et al.’s [31] suggestion that the multivariate adaptive regression splines data mining algorithm had better identification properties than other estimation models. 

The cross-validation results attained by the MARS model yielded 4 terms and third-order interaction. The current findings differed from Faraz et al.’s findings [33], which yielded seven terms with no interaction in the model selection graph. Additionally, Fatih et al. [35] obtained the lowest cross-validation with a MARS model with fifteen terms and third-order interaction. MARS model predicted that bucks were 4.5 kg lighter than does. It further suggests that the influence on the Savanna goat’s body weight was 2.71 when HG was > 100 cm. Therefore, farmers might select goats with more than 100 cm of heart girth during breeding for improving live body weight by about 2.71 kg. The outcomes of the current study might be used by Savanna goat farmers for breed standards in the goat breeding programme. 

## 5. Conclusions

In the current study, it was found that there is a highly positive relationship between live body weight and linear body measurements (body weight, body length, heart girth, rump height and withers height) of Savanna goats. Based on our findings, it was concluded that the MARS algorithm was very successful in estimating the live weight of Savanna goats from linear body measurements. Multivariate adaptive regression splines indicated that sex, body length, body weight, heart girth, rump height and withers height affected the body weight. The Savanna goat farmers can use these findings to improve the live body weight of their goats by choosing in the model the traits which influence the live body weight. The MARS results suggested that sex, withers height, rump height, body length and heart girth might be considered during goat breeding for the improvement of live body weight. In addition, the MARS findings provided information for the linear body measurements of appropriate herd management conditions such as dosage and feeding. This study had limitations and one of them was that only one data mining algorithm was used. Additionally, a small sample size (n = 173) was used. Because of these telling limitations, we recommend that more studies are conducted on the use of MARS and other data mining algorithms using more Savanna goats (n > 173) to assess the effect of linear body measurements on body weight.

## Figures and Tables

**Figure 1 animals-13-01146-f001:**
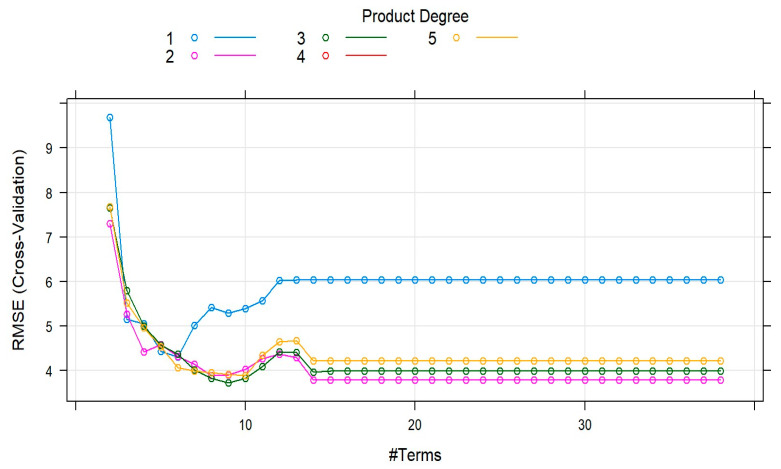
Model selection graph for optimal MARS model.

**Table 1 animals-13-01146-t001:** Descriptive statistics of does.

Traits	N	Mean ± SE	SD	CV (%)	Minimum	Maximum
BW	157	42.62 ± 0.84	10.55	24.76	22.80	66.00
WH	157	65.77 ± 0.42	5.24	7.97	55.00	76.00
RH	157	68.51 ± 0.40	5.02	7.32	59.00	79.00
BL	157	74.59 ± 0.62	7.79	10.41	56.00	99.00
HG	157	83.47 ± 0.78	9.77	11.70	67.00	105.00

BW = body weight, RH = rump height, HG = heart girth, BL = body length, WH = withers height, SD = standard deviation, CV = coefficient of variation, N = number.

**Table 2 animals-13-01146-t002:** Descriptive statistics of bucks.

Traits	N	Mean ± SE	SD	CV	Minimum	Maximum
BW	16	53.30 ± 7.20	13.53	31.01	34.60	102.00
WH	16	67.50 ± 3.54	6.53	9.90	42.00	86.00
RH	16	73.75 ± 2.25	5.68	8.22	65.00	89.00
BL	16	76.623 ± 2.48	7.98	10.67	63.00	93.00
HG	16	86.88 ± 4.06	10.51	12.55	69.00	115.00

BW = body weight, RH = rump height, HG = heart girth, BL = body length, WH = withers height, SD = standard deviation, CV = coefficient of variation, N = number.

**Table 3 animals-13-01146-t003:** Phenotypic correlation of savanna goats, bucks above diagonal and does below diagonal.

Traits	BW	WH	RH	BL	HG
BW		0.74 **	0.99 **	0.90 **	0.94 **
WH	0.80 **		0.64 **	0.89 **	0.71 **
RH	0.71 **	0.86 **		0.85 **	0.93 **
BL	0.72 **	0.71 **	0.61 **		0.89 **
HG	0.88 **	0.71 **	0.69 **	0.65 **	

BW = body weight, BL = body length, RH = rump height, WH = withers height, HG = heart girth, ** = Correlation is significant at the 0.01.

**Table 4 animals-13-01146-t004:** Predictive performances of the MARS model for training and test data sets.

Criterions	Training	Test
Root mean square error (RMSE)	3.541	4.912
Relative root mean square error (RRMSE)	8.186	11.037
Standard deviation ratio (SD_Ratio_)	0.282	0.299
Coefficient of variation (CV)	8.220	10.540
Pearson’s correlation coefficients (r)	0.959	0.961
Performance index (PI)	4.178	5.629
Mean error (ME)	0.000	1.598
Relative approximation error (RAE)	0.006	0.011
Mean absolute percentage error (MAPE)	6.474	7.616
Mean absolute deviation (MAD)	2.650	0.900
Coefficient of determination (Rsq)	0.921	3.576
Adjusted coefficient of determination (ARsq)	0.915	0.877
Akaike’s information criterion (AIC)	331.587	173.987

**Table 5 animals-13-01146-t005:** Multivariate adaptive regression splines algorithm.

BF	Equations	Coefficients
(Intercept)		53.21
BF1	SexM	−4.57
BF2	max (0; WH-63)	0.51
BF3	max (0; 84-BL)	−0.66
BF4	max (0; 100-HG)	−0.46
BF5	max (0; HG-100)	2.71
BF6	max (0; WH-63) * max (0; 75-BL)	0.28
BF7	max (0; 68-WH) * max (0; 100-HG)	0.02
BF8	max (0; 70-RH) * max (0; 100-HG)	−0.02

BF: basic function, Max: maximum, BW = body weight, BL = body length, RH = rump height, WH = withers height, HG = heart girth.

## Data Availability

Please contact the corresponding author (T.L.T.) for the data requests.

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
