# Peer review of "Using Multivariate Adaptive Regression Splines to Estimate the Body Weight of Savanna Goats"

_animals, 2023, doi:10.3390/ani13071146_

Round 1

Reviewer 1 Report

Dear authors, thank you for your submission, the manuscript is interesting but I suggest some integrations in the attached file

Author Response

Dear Editor 

Kindly find the attachment for responding to the reviewers comments.

Reviewer 2 Report

Present paper reports an investigation of the influence of linear body measurements on the body weight of Savanna goats using MARS thought the analysis of 150 Savanna goats.

The paper is complete, well-structured and fully documents the procedure of measurements and statistical analysis of the collected data. It has a set of errors and typos that are noted in the attached file.

The introduction section must contain a paragraph defining the structure of the paper.

Acronyms (BW, BL, HD) are defined several times in the text. 

Despite reporting several times that sick and pregnant animals were excluded from the study, the phases of the gestational cycle in which each of the animals are found are not defined. This could help identify weight variations due to this possible mismatch.

The end of section 2 contains a set of equations derived from the model (MARS) without any explanatory text. They must be removed or explained.

Several paragraphs should be rewritten (Lines 200-206, 216-222), in order to make text easier to read/understand.

Conclusions are too short.

Author Response

Dear Editor 

Kindly find the attachment as the response to the reviewers comments. 

Reviewer 3 Report

The manuscript investigates the influence of linear body measurements on the body weight of Savanna goats using multivariate adaptive regression splines (MARS). It is an interesting paper; however, I found some text mistakes that need correction. Please check the attached file.

In the section Material and Methods, please specify how you get to Figure 1 (Model selection graph for optimal MARS model).

Author Response

Dear Editor 

Kindly find the attachment for the response to the reviewer 3 comments.

Round 2

Reviewer 1 Report

Dear authors, the manuscript was revised however some points are not fully addressed:

the discussion must be improved and deeper

the practical implications are not sufficient reported

limitations of the study are missing

the model is not well described

attached the file with my specific comments

Author Response

Dear reviewer 

Many thanks for the suggestions for improving our manuscript. We addressed all the comments to improve our manuscript. Kindly find the attachment for the response to your comments.

Many thanks. 
